# Testing for Wearability and Reliability of TPU Lamination Method in E-Textiles

**DOI:** 10.3390/s22010156

**Published:** 2021-12-27

**Authors:** Paula Veske, Frederick Bossuyt, Jan Vanfleteren

**Affiliations:** Centre for Microsystems Technology (CMST), Imec and Ghent University, Technologiepark 126, 9052 Gent, Belgium; frederick.bossuyt@ugent.be (F.B.); Jan.Vanfleteren@UGent.be (J.V.)

**Keywords:** e-textiles, stretchable electronics, wearables, packaging, washing reliability

## Abstract

Electronic textiles (e-textiles) and wearable computing have been emerging increasingly during the last decade. Since the market interest and predictions have grown, the research into increasing reliability and durability of wearables and e-textiles is developing rapidly. The washability of different integration methods and resistance to mechanical stress are the main obstacles being tackled. However, the freedom of movement and overall comfort is still often overlooked during the development phase. It is essential to see the e-textile product as a whole and consider several aspects of user experience. This work will focus on developing and improving the thermoplastic polyurethane (TPU) lamination integration method for e-textiles. In the work, a stretchable copper-polyimide based circuit was laminated onto knit fabric using various TPU films and stacks. The study shares measurable characteristics to determine which material assembly and design would ensure the highest durability for the electronics part without losing its original textile softness, flexibility and stretchability.

## 1. Introduction

The wearables and smart textile market have been growing exponentially during the last decade. The following 5–10 years are predicted to be fast-paced in growth by several market reports and funding bodies [1,2,3]. E-textiles’ readiness for the market also depends significantly on the user’s attitude towards wearing and using them. Reliable but uncomfortable garments or vice versa can be unacceptable for the public, as seen in the currently small amount of e-textile products available.

While the smart textiles gathered traction already in the early 1990s, it was the early 21st century when Leah Buechley and Mike Eisenberg collaborated with SparkFun Electronics to develop Lilypad Arduino that could be used for sewing electronics easily on garments and other textile goods [4]. Now, smart textiles (including e-textiles) are most commonly defined as fabrics or garments that have technology that senses and reacts to the environment to which it is exposed. As a result, the wearer will be able to experience “increased functionality” [5]. While “increased functionality” is indeed essential, the durability and the comfort of the product are also aspects that define the product’s popularity.

One of the most researched areas in e-textiles is reliability against domestic washing procedures and finding the most durable solution. While one of the biggest obstacles to doing such research uniformly is lack of standardization for e-textiles washability testing, the most commonly used references are:ISO (International Organization for Standardization) standard “ISO 6330:2012—Textiles—Domestic washing and drying procedures for textile testing”.AATCC (American Association of Textile Chemists and Colorists) standard “AATCC 135—Test Method for Dimensional Changes of Fabrics after Home Laundering”. [6,7].

Various effects due to laundering are being checked in current research in different aspects, e.g., stability of electrical properties (change in resistance, etc.) or stability of the functionality overall [8,9,10,11,12]. One of the techniques for making e-textiles durable is laminating conductive or functional parts with TPU films [13,14,15,16]. This method can be highly efficient by using continuous roll press during manufacturing, which is already available in most textile and garment productions. Thus, considering lamination as an encapsulation method for smart textiles is highly relevant.

However, when incorporating electronics, such as sensors and actuators, in the garments, they need to keep the same or similar level of stretchability, flexibility, softness and sensitivity against the skin—their comfort. Comfort is difficult to define, but it is mainly described as a harmonious state between humans and the environment [17]. However, since personalisation and everyday well-being are a growing trend, “comfort” is also examined through other sensations, such as freedom of movement and psychological and sensorial comfort [18,19,20]. Moreover, comfort in wearables has been researched to some extent, where the relationship between functionality and comfort is often studied [21,22,23]. Knight et al. [24] even developed a “comfort assessment tool” for wearables where by using comfort rating scales (CRS), six aspects are considered: emotion, attachment, harm, perceived change, movement and anxiety. It would be valuable to have some perception of possible (dis)comfort which the end-user can experience wearing e-textiles already during early phases of development. Sensorial discomfort can be ruled out by designing electronic parts not to touch the skin. Thus, it leaves the “freedom of movement” constriction that may be hugely affected by the integration method.

This work focuses on finding the balance between the comfort and reliability of e-textiles. The current research is an extension of a previous study based on laminating a copper-polyimide bus system with different thermoplastic polyurethane (TPU) films and stacks on knit fabrics and conducting reliability (washing) tests on the samples [25]. This study also focuses on increasing comfort (freedom of movement) by higher stretchability and reliability in thinner TPU stacks by reducing the TPU area. Results will present measurable characteristics for freedom of movement based on tensile tests [26]. In addition, additional domestic washing cycles based on ISO 6330-2012 standard were executed for reliability tests [7].

## 2. Materials and Methods

The work consists of two main parts:Wearability: Testing for freedom of movementReliability: Testing for washing reliability

The materials and sample concept always stayed the same. Used materials are listed in Table 1. Overall, the materials were categorized into three types: textile, electronic and integration materials.

**Textile.** Knit fabric from Eurojersey with high elasticity was always used as a textile carrier. Fabric can be typically used for several applications, such as sportswear, underwear and swimwear. Textile swatches’ (used in the tests) longer edge was always cut along the warp thread. For wearability tests, the same size (300 mm × 60 mm) swatches as for integrated samples were cut.

**Electronics.** The electronics consist of a flexible printed 4-track I^2^C bus system with 3 interposer islands, and meander-shaped interconnects in the test system. Copper tracks on the polyimide-copper base sheet were etched with H_2_SO_4_ with standard lithography technique. The circuit was then laminated with ShengYi SF305C coverlay, which had perforations that allowed contact with the copper. Finally, the tracks were cut from the sheet using a CO_2_ laser. The copper-polyimide meander design was chosen based on previous work, proving the best durability and low electrical resistance [25]. The meander design also included an additional copper track at the top and the bottom parts for reinforcement purposes only; these 2 tracks were not electrically connected to the circuit (Figure 1a). For mimicking stiff connections to rigid electronics, printed circuit boards (PCBs) were connected to the two edge interposer islands using SAC305 solder paste and reflown in a vapour phase reflow oven (IBLSLC 300) at 250 °C.

**Integration materials.** For integrating the electronics system onto the textile, two different TPU films were tested. Stretchable Bemis 3914 (thickness 100 μm) and Prochimir TC5011 (thickness 250 μm) TPU films were used in single and stacked versions. TPU was cut in the shape of the meander circuit to increase the freedom of movement. The film area was increased more around the interposer islands and a few millimetres around the meander bus tracks (Figure 1a,b,d).

**Figure 1 sensors-22-00156-f001:**
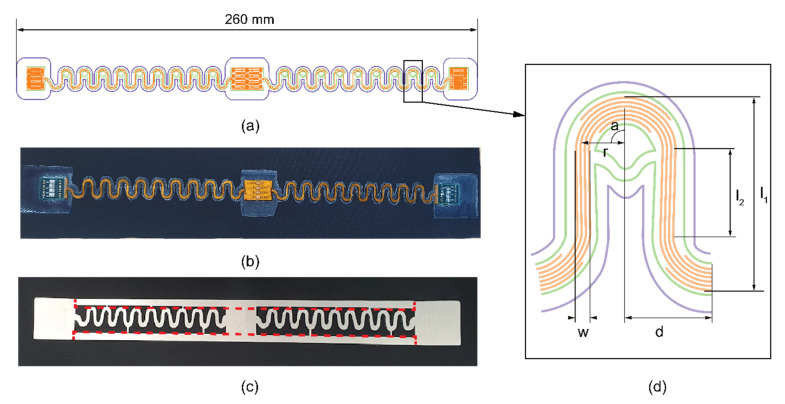
Sample set-up. (**a**) Test batch sample outline: purple line TPU; green line coverlay; orange lines copper. (**b**) Test batch sample design V1 on the Eurojersey swatch fabric sized 300 × 60 mm. (**c**) TPU meander on white release paper with the frame. The red dotted line indicates where the frame is cut off before lamination on the fabric. (**d**) Close-up of the meander shape and measurements (see Table 2).

Four different versions of sample batches were prepared for each test (Table 3). Every batch consisted of 5 identical samples. Thus, 20 samples were prepared for both wearability and reliability test. The overall process for sample batches preparation was:Preparation of electronics and stretchable copper-polyimide meander system;Preparation of fabric swatches and TPU film layers in the shape of the copper-polyimide meander system;Sandwiching the copper-polyimide meander system between the TPU layers in 4 versions (Table 3);Laminating the sandwiched copper-polyimide meander system onto the knit fabric swatches.

The first two steps (preparation of electronic circuit, fabric swatches and TPU) is explained above with material descriptions. In the third step, the TPU film layers were used to sandwich the copper-polyimide stretchable circuit for encapsulation and laminate onto the knit fabric. Since the TPU was cut in the shape of the meander interconnects (Figure 1), the handling of thinner 100 μm Bemis TPU film was more complex than the thicker Prochimir film. Thus, a frame around the meander circuit was created (Figure 1c). The frame was cut off before laminating the sandwiched (encapsulated) circuit onto the fabric swatch, as shown by the red dotted line in Figure 1c.

Table 3 indicates the entire thickness of TPU layers (on the front and backside of the electronic circuit) before the lamination process and after. The last TPU thickness indicates how different the films’ melting flow was, which is also reflected in the stretchability in the later stages. The lamination parameters used were 170 °C with 4 bar pressure for 40 s. Lamination was done on a Hotronix^®^ Air Fusion IQ^®^ heat press where the heating plate is on the top and the bottom plate is at room temperature.

**Wearability test method.** The wearability or the freedom of movement test was based on ISO 20932-1:2018 standard for determining fabrics elasticity with a strip method [26]. Each sample with a laminated circuit was clamped to the Lloyd Instruments LS5 machine from the TPU area (not only from the textile) as seen on Figure 2.

The samples were stretched 20% of their length, held at the stretched position for 10 s and released (Figure 3). Each sample was stretched 100 cycles. Since strain was fixed, the primary measurement analysed was the load in different cycles and moments during the cycles. The bus system’s functionality was checked with a multimeter before and after the test.

**Reliability to domestic washing procedure method.** Washability of samples was tested based on ISO 6330-2012 standard that defines domestic washing and drying procedures [7]. Table 4 gives an overview of the exact washing program used during the tests. Two kilograms of microfiber towels (Nabaji, 88% PES, 12% PA) were included in each washing cycle. The washing tests were performed in an Electrolux W465H professional washing machine. The samples were washed all together in a mesh washing bag.

The tests aimed to reach 25 washing cycles with no errors in the sample batch. All samples were washed simultaneously. In the previous tests [25], using textile-based circuits with I^2^C, we had seen issues that the rise time of the I^2^C blockwaves is too slow. It was solved by tuning the pull-up resistors. As the rise time is depending on the resistance of the lines and the capacitance between them, we believe that due to washing the I2C signal quality is mainly affected by the resistance change. Furthermore, if the resistance of the tracks becomes too high, the voltage drop on the line will be too high. Sensors on the bus will not reach their necessary supply voltage and not function properly. Therefore, he each track’s resistance was checked after every washing cycle with a multimeter. Following five successful washing cycles, measurements were taken every two washing cycles.

## 3. Results

Results were analysed in two parts as mentioned in the Materials and Methods section’s beginning:Wearability: Testing for freedom of movementReliability: Testing for washing reliability

Firstly, it was observed, the melting areas of stacks and TPUs differed, as seen in Table 3 last two columns. In V4, the thickness of the TPU area changed from 1000 µm to 750 µm, whereas in V1, it changed from 200 µm to 170 µm. Figure 4 presents V1 and V4 samples after lamination and tensile tests. The melting area of the TPU is significantly larger in V4 compared with V1. The circled areas on the figure show higher stress areas where TPU delaminated from the textile at the TPU edges and/or the copper-polyimide meanders broke mostly.

**Wearability.** Testing for freedom of movement, series on tensile tests was conducted based on ISO 20932-1:2018 standard (Figure 3) [26]. The tests compared plain knit fabric swatches to the same sizes swatch samples with integrated electronic circuits. Most of the measurement points were taken at the test cycles’ midpoint (at 12% strain) (Figure 5).

For comparison, three characteristics were considered: Young’s modulus (*E*), Stress decay due to time (σ_t_), and Work-in-Moment (WiM). All the measurements showed are the average of 5 samples in the sample batch (see Figure 5 and Figure 6). 

1.Young’s modulus (*E*), measuring and comparing the elasticity between different samples at the midpoint (12% strain point).

Based on the tensile tests’ data, strain (*ε*), stress (*σ*) and Young’s modulus (*E*) were calculated:E=FAlΔl0=σε,
where
*F* is the force exerted on the sample during the tensile test, measured in Newton (N),*A* is the actual cross-sectional area in mm^2^,*l*_Δ_ is the change in length in mm,*l*_0_ is the original length of the sample in mm,*ε* is strain,*σ* is uniaxial stress in kPa.

The Young’s modulus was calculated based on the tensile test last cycle values at the strain’s midpoint. Figure 6a presents Young’s modulus comparison of various stacks in test samples and the plain knit fabric (last column). It was seen that the V1 stack on the knit fabric and the plain fabric have very similar values. Thus, the V1 stack has very little or no effect on the stretchability compared to the plain knit fabric.

Contrary to V1, V4 Young’s modulus is over five times larger, making the textile extremely stiff. The melted TPU stack used in V4 is also 4 times thicker than in V1, which affects the stretchability and elasticity of the whole sample. The TPU thickness also means that the melting area is larger, and the meander-shaped TPU melts more together than V1 or V2 (see also Figure 4, Table 3).

2.Stress decay due time (*σ_t_*, %), measuring the maximum load of the last cycle before and after the 10 s holding period [26].

The goal was to show how much the force changed over the holding period, indicating the ageing of the sample and how elastic or plastic it is. Latter can be additionally defined by measuring the sample length before and after the test (while maintaining a constant load) to check the change in the sample length.

Stress decay due time (*σ_t_*, %) was calculated as follows:$$\sigma_{t}=\frac{L_{maxb}-L_{maxa}}{L_{maxb}}\times 100\%,$$
where
*L_maxb_* is the maximum load of the last cycle before the 10 s holding period,*L_maxa_* is the maximum load of the last cycle after the 10 s holding period.

Figure 6b exemplifies stress decay after the 10 s holding period in the last cycle of the tensile test. Results indicate that although V1 and V2 have almost twice the stress decay (15–17%) compared to the plain knit fabric, V3 and V4 stress decay over the holding period is even larger and amounts to around 23%. It aligns with Young’s modulus graph by showing that more elastic samples (requiring less force to elongate them) also have less force decay and can keep their original shape for a longer time.

Moreover, it was observed that the force decay after some layering did not change, as seen in V3 and V4, showing similar results while V1 and V2 still had a difference. The question should be investigated more deeply to see from which layering onwards decay difference ceases to matter.

3.Work-in-Moment (WiM, %), measuring and comparing the hysteresis at the midpoint (12% strain point) between different cycles and checking how fast and/or intensively it changes.

*WiM* enables the comparison of hysteresis of different samples and cycles. The main aim of *WiM* was to see how long it takes each stack to stabilize the hysteresis loop and keep the area similar to coming stretch cycles.

Work-in-moment (*WiM*, %) was calculated as follows:$$WiM=\frac{\left(\sigma_{1}-\sigma_{2}\right)}{\sigma_{1}}\times 100\%,$$
where
*σ*_1_ is the maximum load of the first cycle in comparison, at the midpoint,*σ*_2_ is the maximum load of the second cycle in comparison, at the midpoint.

Figure 6c shows the *WiM* difference between the first and hundredth (last) tensile test cycle.

Thus, the results obtained to the chart were calculated as follows:(1)$$WiM_{1st\;cycle}-WiM_{100th\;cycle,}$$

That would indicate how stable and elastic each sample stack is compared to plain fabric. While V1 and V2 hysteresis loop difference in the plain fabric is 2–3 times larger, V3 and V4 are 5 times larger. It also shows how the hysteresis loop size was also greatly different between thicker and thinner samples. The area of each loop was considerably different, although it was observed that the proportionally the loop stabilized similarly,

The graph also aligns with the stress decay results, indicating how V1 is more elastic. Moreover, after a certain layer thickness and melting area, as in samples V3 and V4, the plastic properties stay the same, and hysteresis loop changes stay similar (as it was also in stress decay results).

Figure 7 displays how plastic and inelastic the V4 sample is, while the V1 sample shows little to no changes in shape after 100 cycles of 20% elongation. It was seen how the larger melting area of thicker TPU stacks on V3 and V4 create considerably more restriction and plastic properties and behaviour.

**Reliability.** Reliability to domestic washing procedure was tested based on ISO 6330-2012 standard (Table 4) [7]. The main goal was to complete 25 washing cycles without introducing errors in the circuit. After every wash cycle, the circuit functionality was evaluated using a multimeter. Electrical continuity of the Cu metal tracks was measured. After reaching five successful washing cycles, measurements were taken every two washing cycles.

The resistance of each bus system interconnect track stayed stable until failure or end of tests, being on average 16.5 ohm/m (non-connected additional tracks were not measured). Figure 8 summarises reliability measurement results under the washing tests. V1 sample set had no failures in the sample set and reached 25 washing cycles indicating that the TPU thickness does not necessarily need to be proportional to the level of protection (Figure 8a). In this way, the thinnest stack of Version 1 facilitates successful integration by giving freedom of movement to the entire system and relieving possible buckling. Meanwhile, the V4 sample set had first failures after 5 test cycles, and none of the samples reached 25 cycles (Figure 8).

The copper-polyimide samples buckled and cracked in certain areas due to the stiff TPU only covering part of the textile swatch. Failures in tracks seemed to occur around interposer islands where the circuit transitioned into meanders, indicating a high-stress area and the main point of failure caused by mechanical strain and buckling (Figure 9). The buckling on the thicker stacks (V3, V4) was visible during tensile tests, as seen in Figure 10. In order to minimize the stress, the circuit wants to buckle because of elongation. Buckling can occur with thinner TPU, but thicker and stiffer TPU prevents this, and there is localized stress build-up in the circuit and crack generation in the Cu. Thus, TPU with a thinner and softer layer is the better option for comfort while still providing adequate protection against external factors, like wear and tear and humidity.

Overall, it was seen that stiffer and thicker TPU stack limited the sample movement and created specific stress areas, as seen in Figure 4. However, that contradicts previous results where thicker TPU stacks provided additional durability to samples [25].

## 4. Discussion

**Wearability.** As discussed in the beginning, wearability is a broad concept. It is still often tested mainly with subjective user tests which need accurate questionnaires and interpretation. The current study based the wearability test on the tensile test and examined it mainly on comparing the plain knit fabric versus knit fabric with an integrated electronic circuit based on three characteristics: Young’s modulus, stress decay due time and Work-in-Moment.

It was clear that the V3 and V4 samples are much more plastic than the V1 and V2 based on samples’ shape and length after the tests (Figure 7) and Young’s modulus results (Figure 6a). V1 and the plain fabric had very similar Young’s modulus results, showing that the stretchability of either can be interchangeable. A subjective users’ study is typically conducted to determine comfort level, and these results may be used to strengthen or complement the study.

The decay of the force changes in steps—7% comparing the plain knit fabric to the V1 and 8% comparing V1 to V4. It was observed that the TPU layering increases the force decay, but the more elastic samples (V1, V2) still had considerably lower deterioration. WiM graph also aligns with the stress decay results and show how the V1 stack gives the most elastic version. It was observed that most plastic stacks (V3, V4) had the same results in both cases, indicating how beyond a specific TPU melting area and thickness, the plastic properties stay very similar.

Figure 10a displays how tensile tests brought out visible buckling on the thicker stacks. V4 meanders warped under stress and showed possible delamination points (light areas in Figure 10a). However, the V1 stack visibly had no delamination or extra buckling out of the fabric plane, and the whole laminated system moved with the knit fabric (Figure 10b). Thus, using thinner TPU layers creates no specific stress areas on the circuit, resulting in the absence of failures.

Plastic properties and fast ageing on most clothing are not accepted. Thus, these measurable properties could also additionally support the possible comfort level of the future garment. Still, it needs to be kept in mind that e-textiles application areas vary tremendously. High stiffness that could also offer support or additional reinforcement in certain areas on the body can be used in specific use cases, e.g., in work or medical clothing.

**Reliability.** Stretchable electronics encapsulated with various TPU layering on textiles and their durability to washing procedures were also tested in previous work [25]. The double-layered TPU with the same meander design was more successful in the previous study. However, the prior work did not test meanders with non-connected extra copper together with one layer of TPU in the meander width nor used the meander-shape structured TPU. Figure 11 displays the different layering and TPU area methods in the two studies. In the earlier work, it was apparent that the increased number of layers of TPU (either 100 µm or 250 µm thick) was one of the factors creating more durable integration.

Reviewing the earlier samples (Figure 11b), it appears that having a uniform, non-patterned TPU layer in the TPU area on the whole sample decreases the stress areas around the weakest points observed around interposer and meander junction in any stacks version. While the V4 samples melting area in the current study is more extensive, it still does not cover most of the textile and creates a lot of stress around interposer joints (Figure 4 and Figure 9).

Figure 11b also demonstrates how a larger TPU cover protects the additional smaller TPU area in the previous work tests samples. In this way, it was seen that the copper-polyimide system is less prone to buckling and cracking. Going back to visible buckling during the current tensile tests, it was clear how a reduced yet highly stiff laminated area could not avoid buckling in V3 and V4 samples (Figure 10a).

Presumably, unifying the TPU area creates fewer stress areas. However, it appears that if the TPU layers are minor in surface area and/or thin enough, the uniformity of the whole sample is not disturbed. Thus, the elasticity (cf Figure 6a) stays similar or the same as without the TPU layers. Nevertheless, when the TPU was used to cover the entire sample without shaping/patterning the covered area in any way, the stretchability and elasticity were severely hindered. Figure 12 last 3 bars display Young’s modulus difference in previous samples compared to the plain knit fabric. The least stiff sample (with lowest thickness TPU used, Young’s modulus 9 kPa) still does not emulate the stiffest samples (V4) (Young’s modulus 5 kPa, see Figure 12) in current work.

It is essential to note that the TPU cover area needs to be considered based on application needs, as mentioned before. Washing reliability compared to the freedom of movement can be more important in certain fields and vice versa. The 250 µm TPU melting area was considerably larger in any case. Thus, it would be useful to determine if it can be precisely shaped to cover very specific areas and, by that, gain more flexibility.

Moreover, it is important to note the possible misalignment issues that may happen during the lamination process. Figure 4 lower photo shows the small misalignment around the interposer. The deformations during the lamination process and TPU geometry with the material handling process might create difficulties affecting the final reliabilty and wearability. The possible solutions to avoid the possible misalignment need to be researched further, together with efficient fabrication methods for larger volumes manufacturing. For example, laser-cutting the TPU together with meander circuits needs to be tested.

## 5. Conclusions

This study tested four different TPU stacks on copper-polyimide meanders for the e-textile integration method. The aim was to research each stack’s wearability (freedom of movement) and domestic washing reliability.

Since the thinnest stack (V1) and plain knit fabric elastic modulus were compatible, future work can include comparative end-user studies. It was seen how the thinnest stack had the most similar properties to plain fabric, which exhibited an increased level of comfort and freedom of movement. A subjective end-user study could indicate if the comfort level stays still the same with or without the circuit lamination.

Interestingly, the thinnest stack was also most durable in washing tests. It was seen how increased freedom of movement also decreased specific stress areas that were obvious in thicker stacks. Moreover, the thicker TPU melting areas were considerably larger than in the case of thinner TPU stacks, thus reducing flexibility and enhancing the crack formation around interposer and meander tracks’ transition areas.

A further study could assess the effects of an additional knit fabric layer on top of the circuit. So far, the research has used only TPU layers in the integration process. However, the additional fabric layer would most probably hinder the stretchability but also increase durability and aesthetics. Still, the additional fabric layer on top could help prolong the product’s overall lifetime since delamination of TPU for repairs and reuse of parts could be done more easily and efficiently. Moreover, a more efficient and accurate lamination method on top of the meander circuits will be researched and examined.

## Figures and Tables

**Figure 2 sensors-22-00156-f002:**
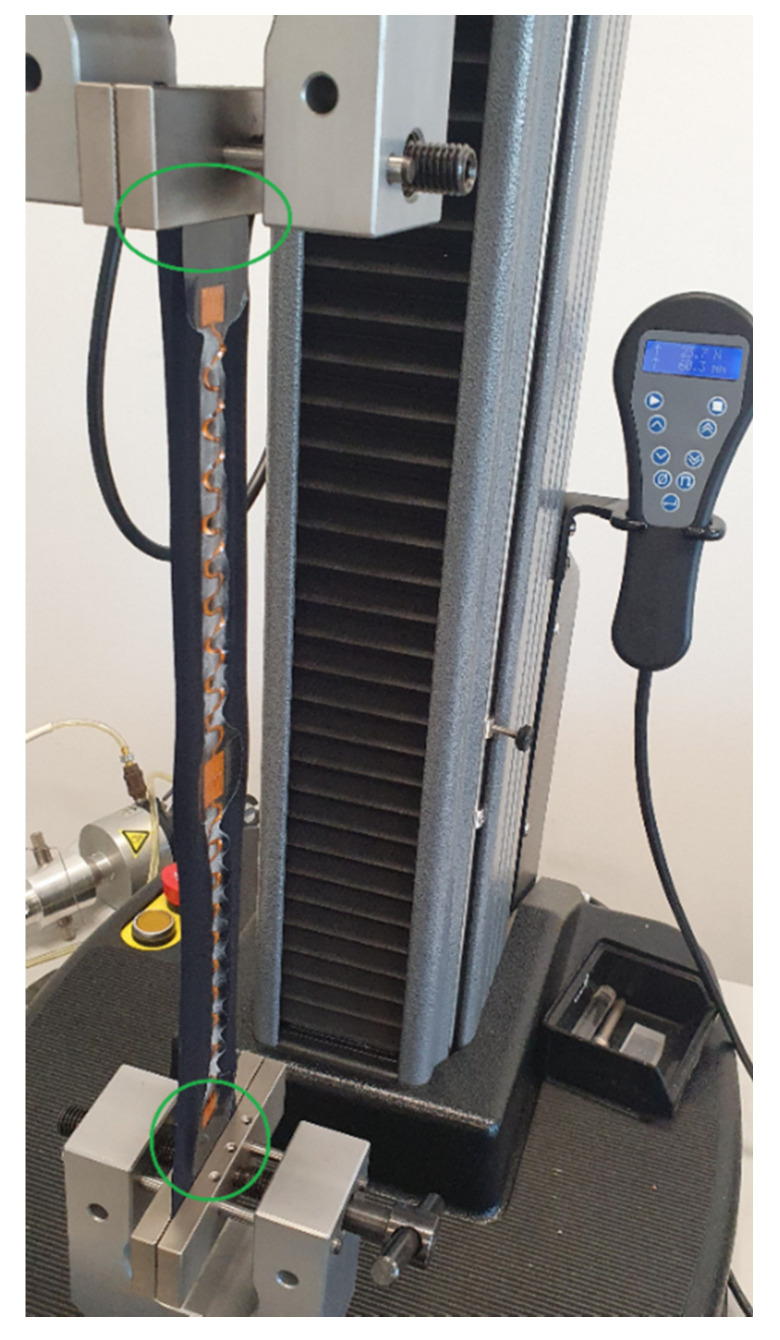
Stretched sample where the green circled areas highlight that the sample was clamped from the laminated fabric area and not just fabric.

**Figure 3 sensors-22-00156-f003:**
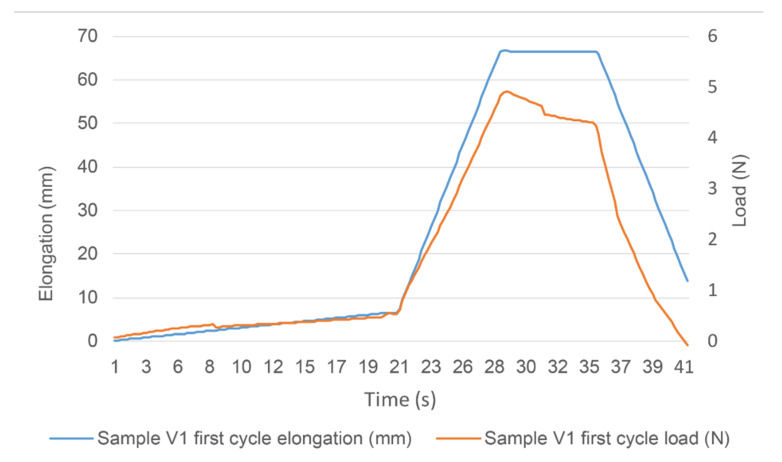
The tensile test one cycle overview, repeated 100 times.

**Figure 4 sensors-22-00156-f004:**
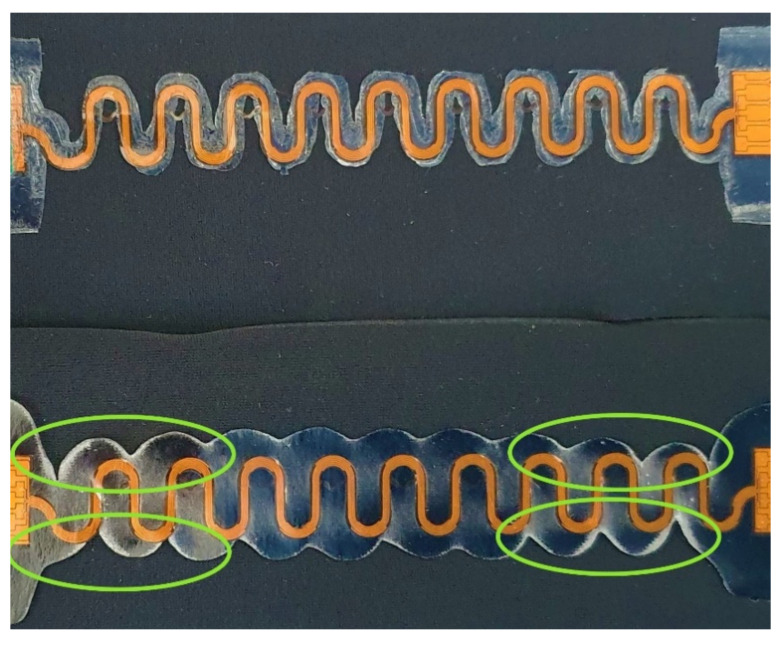
Melting area difference of different versions. Up: Version 1 after lamination and tensile tests. Down: Version 4 after lamination and tensile tests. Circled areas indicate higher stress areas where TPU delaminated from the edges and/or the copper-polyimide meanders broke.

**Figure 5 sensors-22-00156-f005:**
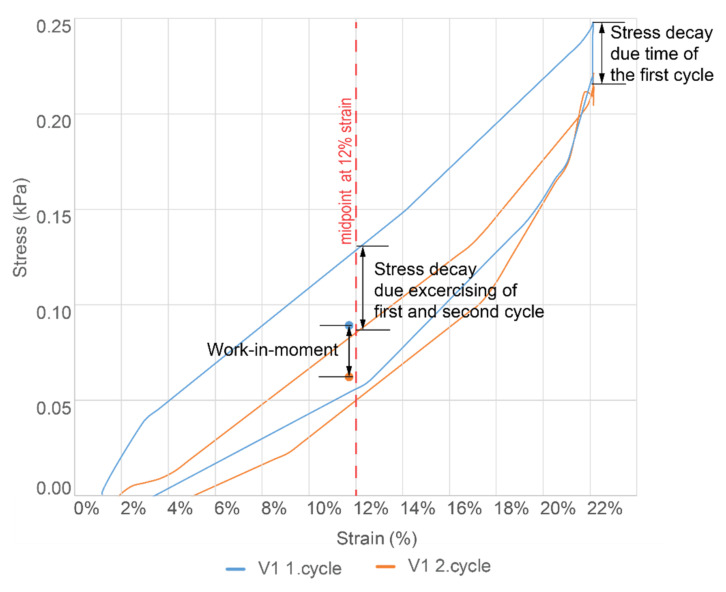
Stress-Strain curve example of the first two cycles from V1, indicating the midpoint position at strain 12%, which was used as the measuring point for determining Young’s modulus (E) and Work-in-moment (WiM). Three characteristics and where they are measured is also shown.

**Figure 6 sensors-22-00156-f006:**
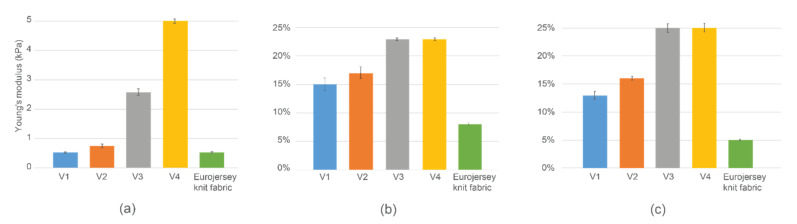
(**a**) Young’s modulus comparison. (**b**) Stress decay after 10 s holding period in 100th tensile test cycle. (**c**) WiM difference between first and last tensile test cycle.

**Figure 7 sensors-22-00156-f007:**
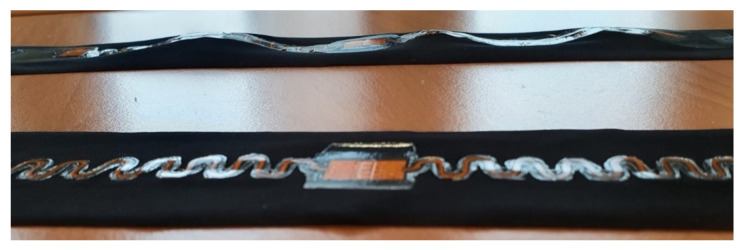
Elasticity difference of V1 and V4 samples. Up: V4 sample after 100 tensile test cycles. The laminated circuit is distorted together with the fabric. Down: V1 sample after 100 tensile test cycles. No distortions into the lamianated circuit or fabric were seen.

**Figure 8 sensors-22-00156-f008:**
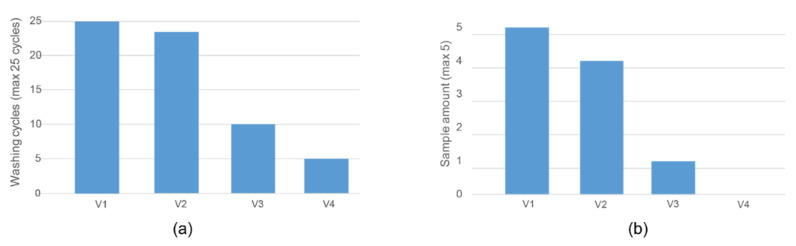
(**a**) First failure appearance during washing tests with test samples. (**b**) Amount of samples per the set of 5 that had no failures after 25 washing cycles.

**Figure 9 sensors-22-00156-f009:**
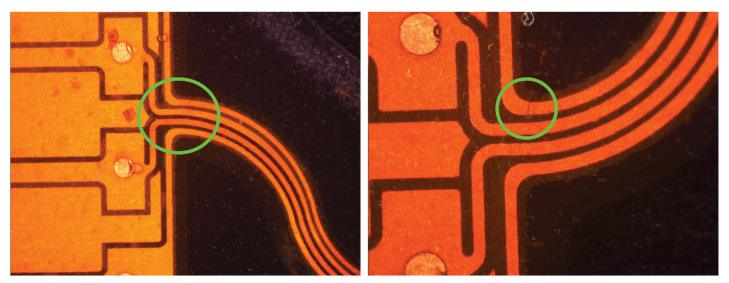
V4 test batch samples with microcracks after washing tests.

**Figure 10 sensors-22-00156-f010:**
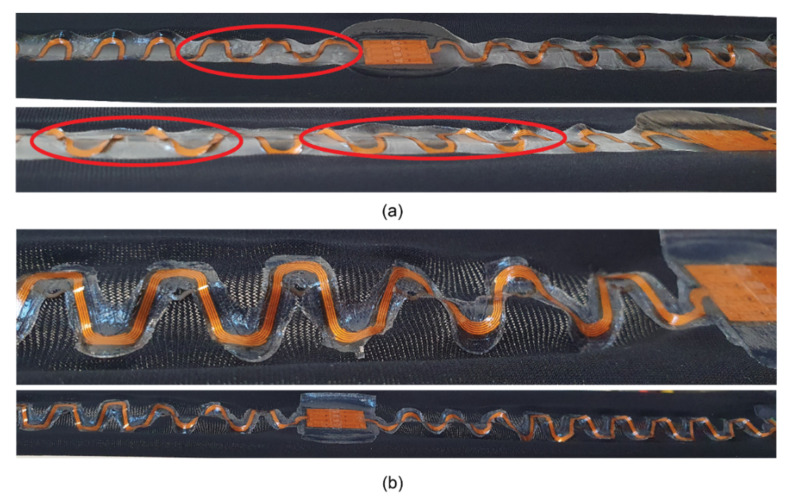
(**a**) V4 under 20% stretch, circled areas highlight buckling. (**b**) V1 under 20% stretch, the laminated circuit buckles together with the fabric but does not release or delaminate from it.

**Figure 11 sensors-22-00156-f011:**
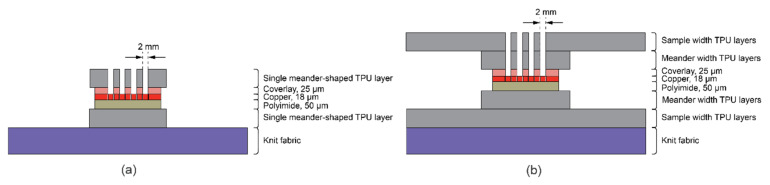
Layering system of samples. (**a**) V1 and V3 layering systems in the current tests. (**b**) Most successful TPU layering system in the previous work, reproduced with permission [25].

**Figure 12 sensors-22-00156-f012:**
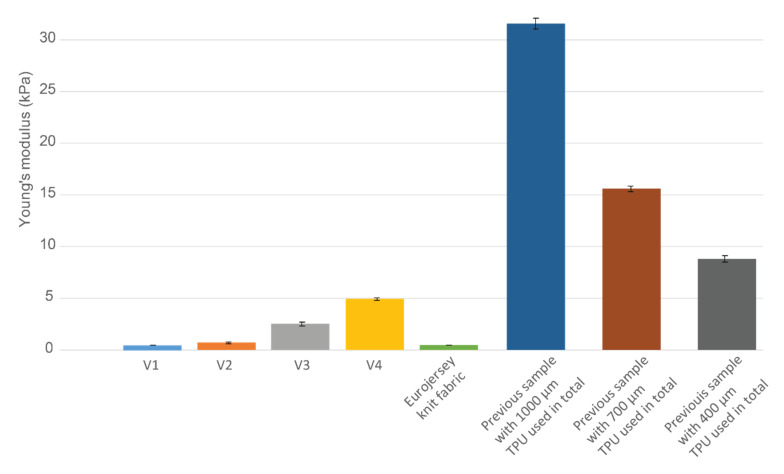
Young’s modulus comparison of current samples, plain knit fabric and previous work samples.

**Table 1 sensors-22-00156-t001:** Materials overview.

Material	Dimensions	Characteristics
Eurojersey Sensitive^®^ Fit (68% microfibre polyamide, 32% elastane)	Weight 213 g/m^2^, sample size 300 × 60 mm	Knit fabric
Etched copper-polyimide 4-track meander bus system	18 μm Cu, 50 μm PI (polyimide film)	
Polyimide based Coverlay film ShengYi SF305C	25 μm thickness	
Printed circuit board (PCB) with 4 gold-plated connection pads	14 × 12.5 × 0.2 mm	
SAC305 solder paste—96.5% tin, 3% silver, and 0.5% copper		
TPU film Bemis 3914	100 μm thickness	E = 2.2 Mpa,
		glue line temp 120–170 °C
TPU film Prochimir TC5011	250 μm thickness	E = 28 MPa,
		glue line temp 140–170 °C

**Table 2 sensors-22-00156-t002:** Meander dimensions based on Figure 1d.

Mark	Description	Measurement	Unit
l_1_	Width of the meander bus	11.4	mm
I_2_	The length of the straight part connecting the top and bottom arc	4.8	mm
w	Total width of meander including 4 Cu tracks	0.9	mm
d	Half of the period between the middle top and middle bottom part	5	mm
r	Meander radius	2.45	mm
a	Meander angle	90	°
	Spacing between copper tracks	0.1	mm
	Copper track width	150	μm

**Table 3 sensors-22-00156-t003:** Samples overview.

Sample Name	TPU Used	Total TPU Thickness before Lamination, μm	Total TPU Thickness after Lamination, μm
Version 1, V1	Bemis 3914	200	172
Version 2, V2	Bemis 3914	400	387
Version 3, V3	Prochimir TC5011	500	395
Version 4, V4	Prochimir TC5011	1000	747

**Table 4 sensors-22-00156-t004:** Washing test procedure overview based on ISO 6330-2012 standard.

Nr	Washing Phase	Time, min	Temperature, °C	Spin, Rmp (Revolutions per Minute)	Water Volume, l
1	Main wash, detergent nr 3	15	40	49	15
2	Rinse 1	3	Coldwater	49	19
3	Drain 1	8	Coldwater	49	-
4	Rinse 2	3	Coldwater	49	19
5	Drain 2	8	Coldwater	49	-
6	Rinse 3	3	Coldwater	49	19
7	Drain 3	8	Coldwater	49	-
8	Rinse 4	3	Coldwater	49	19
9	Drain 4	8	Coldwater	49	-
10	Spin	5	Coldwater	1100	-

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
