# Peer review of "Testing for Wearability and Reliability of TPU Lamination Method in E-Textiles"

_sensors, 2021, doi:10.3390/s22010156_

Round 1

Reviewer 1 Report

In the manuscript by Veske et. al., systematic evaluations on the wearability and reliability have been reported for e-textiles based on TPU lamination method. A stretchable copper-polyimide circuit was laminated on the textiles by using several different TPU films. The measurements and analyses shed light on the optimal lamination approach to preserve the freedom of movements and the washing reliability. As a counterintuitive take-home message, a thin TPU film may offer the most durable protection against external mechanical strains and washing tests. I recommend the manuscript to be considered for publication after addressing the following technical issues:

  • In the bottom image of Figure 4, the TPU layer is notably misaligned with the copper meander. The issue may alter the stress distribution on the laminate during mechanical deformations/washability tests. In addition, the origin for the misalignment should be discussed (e.g. errors in manual manipulations, distortions during the lamination process, compliant nature of the TPU film in meander geometry).
  • It is rather difficult to tell whether the circuit is delaminated from the textile or bent together with the textile in the top image of Figure 7.
  • The lamination temperature is much higher than the typical temperature tolerance of regular textiles. Are there any negative impacts on the properties of the textile? How to mitigate these effects?     

Author Response

The authors would like to give thanks for the notes.

Figure 4 misalignment issue was indeed not discussed. The discussion for it was added at the end of the "Discussion" part (lines 385-392) and "Conclusion" (lines 412-413)

Figure 7 caption was updated with further explanations.

There were no negative impacts seen for the textile after the lamination. However, it is important to note it highly depends on each textile used. Thus, if the textile is changed, it needs to be considered. Moreover, the TPU choice can be reconsidered based on textile choice, since the range of TPUs is quite big.

Reviewer 2 Report

Dear Author,

The paper it is interesting, however according to my point of view, important results are missing.

The authors explain the electronic consist of I2C bus. However, only the electrical conductance it is evaluated. In my opinion, the propagation  properties of the signal integrity  must be evaluated (by means of eye diagram or impedance model).   

Just for your reference, you can check DOI: 10.1080/09205071.2015.1106987

Author Response

The authors would like to thank for the feedback!

For our textile-based circuits using I2C, we have seen issues that the rise time of the I2C block waves is too slow. By tuning the pull-up resistors, this can be solved. As the rise time depends on the resistance of the lines and the capacitance between them, we can state that due to washing, the I2C signal quality is mainly affected by the resistance change (what was investigated). The capacitance will be more or less the same as it is dependent on the materials used.

We hope this is a sufficient explanation of why we did not investigate signal integrity and evaluate it by impedance model in the current work. Still, it can definitely be part of future research.

Reviewer 3 Report

The manuscript has focused on developing and improving the thermoplastic polyurethane lamination integration method for e-textiles. The manuscript describes  measurable characteristics to determine which material assembly and design would ensure the highest durability for the electronics part without losing its original textile softness, flexibility and stretchability. The topic is interesting and the manuscripts are well written. The introduction provide sufficient background and include  relevant references.The research design is appropriate and the conclusions supported by the results. The figures are well designed. 

It needs to be corrected:

Error! Reference source not found - line 96, 115,164
Pleas explain what PI mean (Table 1).

Author Response

The authors would like to thank for the notes.

The Error message has been corrected. PI has been explained.

Round 2

Reviewer 2 Report

The justification for measuring only the resistance line and not the impedance or signal integrity should be included in the article.

Author Response

Thank you for the notes. 

The authors added the additional explanation to rows 167-182.